# Kinesin and dynein use distinct mechanisms to bypass obstacles

Luke S Ferro[1], Sinan Can[2], Meghan A Turner[3], Mohamed M ElShenawy[1], Ahmet Yildiz[1,2,3]*

[1]Department of Molecular and Cell Biology, University of California, Berkeley, Berkeley, United States; [2]Department of Physics, University of California, Berkeley, United States; [3]Biophysics Graduate Group, University of California, Berkeley, Berkeley, United States

**Abstract** Kinesin-1 and cytoplasmic dynein are microtubule (MT) motors that transport intracellular cargoes. It remains unclear how these motors move along MTs densely coated with obstacles of various sizes in the cytoplasm. Here, we tested the ability of single and multiple motors to bypass synthetic obstacles on MTs in vitro. Contrary to previous reports, we found that single mammalian dynein is highly capable of bypassing obstacles. Single human kinesin-1 motors fail to avoid obstacles, consistent with their inability to take sideways steps on to neighboring MT protofilaments. Kinesins overcome this limitation when working in teams, bypassing obstacles as effectively as multiple dyneins. Cargos driven by multiple kinesins or dyneins are also capable of rotating around the MT to bypass large obstacles. These results suggest that multiplicity of motors is required not only for transporting cargos over long distances and generating higher forces, but also for maneuvering cargos on obstacle-coated MT surfaces.

DOI: https://doi.org/10.7554/eLife.48629.001

## Introduction

Kinesin and dynein move towards the plus- and minus-ends of MTs, respectively, and play major roles in intracellular cargo transport, cell locomotion, and division (*Reck-Peterson et al., 2018*; *Verhey et al., 2011*). Although these motors have complementary functions on MTs, they have distinct structural and mechanistic features. Kinesin-1 contains a globular motor domain that binds the MT and hydrolyzes ATP. Two identical motor domains are connected by a short neck-linker to a common tail (*Figure 1A*) (*Verhey et al., 2011*). In vitro studies have shown that kinesin moves by coordinated stepping of its motor domains, in a manner akin to human walking (*Yildiz et al., 2004*; *Asbury et al., 2003*). It follows a single protofilament track on the MT and almost exclusively steps forward without frequent sideways or backward motion (*Ray et al., 1993*; *Can et al., 2014*). Unlike kinesin, dynein's motor domains are large heterohexameric rings of AAA+ ATPase subunits that connect to the MT through a coiled-coil stalk (*Figure 1A*) (*Roberts et al., 2013*). Stepping of the dynein motor domains is not tightly coordinated (*DeWitt et al., 2012*; *Qiu et al., 2012*). Instead, either monomer can take a step while the other serves as an MT tether (*DeWitt et al., 2012*; *Reck-Peterson et al., 2018*). Dynein has a large diffusional component in its stepping behavior, resulting in frequent sideways and backward steps (*DeWitt et al., 2012*). Differences in the stepping behaviors between these motors may influence their cellular functions (*Hancock, 2014*).

Intracellular transport takes place in a highly crowded and dynamic cytoplasm. The MT network is densely decorated with obstacles such as MT-associated proteins (MAPs), stationary organelles, protein aggregates, MT defects, opposing motor traffic and other cytoskeletal filaments (*Dixit et al., 2008*; *Che et al., 2016*; *Liang et al., 2016*). It is not well understood how motors transport cargos efficiently throughout the cell despite these challenges. Previous in vitro studies suggested that

*For correspondence:
yildiz@berkeley.edu

Competing interests: The authors declare that no competing interests exist.

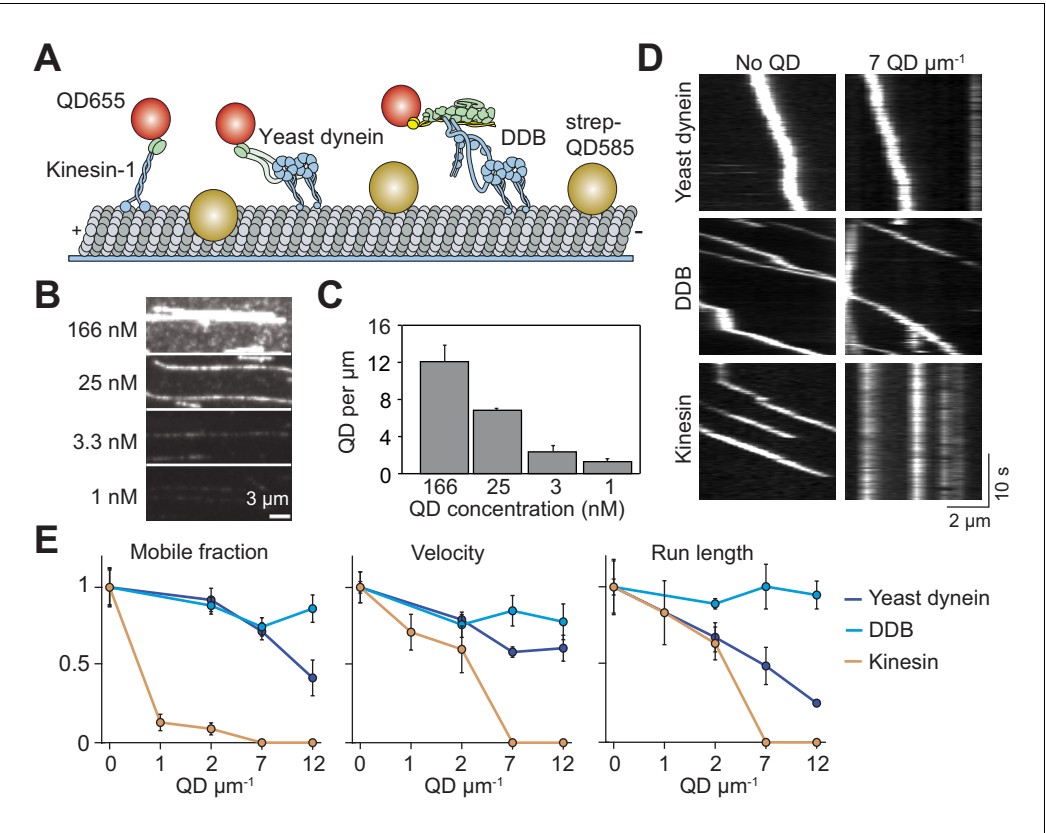

**Figure 1.** Single dynein motors, but not kinesin motors, bypass QD obstacles. (**A**) Schematic of single-molecule motility assays on surface-immobilized MTs decorated with streptavidin-coated QD585 obstacles. Human kinesin-1, yeast dynein, and mammalian DDB are labeled with QD655 at their tail domain. (**B**) Example fluorescent images of QD585 obstacles on MTs at different QD concentrations. (**C**) The linear density of QDs on MTs at different QD concentrations (mean ± SD, from left to right $n$ = 97, 98, 104 and 90 MTs from two technical replicates). (**D**) Kymographs show the motility of QD655-labeled motors on MTs with or without QD obstacles. The QD585 signal is not shown. (**E**) Mobile fraction, velocity and run length for all three motors were all normalized to the no QD condition (mean ± SD, three independent experiments). Run-length values represent decay constants derived from a single exponential decay fit. From left to right, $n$ = 271, 423, 405 for kinesin, 315, 407, 197, 168 for yeast dynein, and 636, 502, 356, 509 for DDB.

DOI: https://doi.org/10.7554/eLife.48629.002

The following figure supplement is available for figure 1:

**Figure supplement 1.** Analysis of single-molecule trajectories on surface-immobilized MTs without normalization.
DOI: https://doi.org/10.7554/eLife.48629.003

motors need to take sideways steps to avoid obstacles on their path (*Schneider et al., 2015*). In agreement with this idea, kinesin motility is strongly inhibited by obstacles such as catalytically-inactive motors, MAPs or cell extract (*Schneider et al., 2015*; *Dixit et al., 2008*; *Telley et al., 2009*). While these motors can occasionally bypass obstacles by detaching and reattaching to neighboring protofilaments, they stall or detach from the MT in most cases (*Schneider et al., 2015*; *Leduc et al., 2012*). Kinesin-2, a different kinesin family member with faster MT detachment/reattachment kinetics (*Feng et al., 2018*) and increased side-stepping ability (*Hoeprich et al., 2014*), bypasses obstacles more successfully than kinesin-1. Dynein was expected to be less sensitive to obstacles than kinesin because of its elongated structure and frequent sideways stepping (*DeWitt et al., 2012*). Yet, in vitro studies on isolated mammalian dynein observed that the motor reverses direction when encountering a MAP obstacle rather than bypassing it (*Dixit et al., 2008*; *Soundararajan and Bullock, 2014*). However, these studies were conducted before it was understood that mammalian dynein alone is autoinhibited and its activation requires assembly with dynactin and a cargo adaptor (*Schlager et al., 2014*; *McKenney et al., 2014*; *Trokter et al., 2012*). Therefore, how active dynein

motors bypass obstacles on MTs is not well understood (*Ruensern Tan et al., 2019*; *Elshenawy et al., 2019*).

Unlike helicases, unfoldases and DNA/RNA polymerases, which usually function as individual motors (*Singleton et al., 2007*), cytoskeletal motors often operate in teams to transport a cargo (*Encalada et al., 2011*; *Rai et al., 2016*; *Levi et al., 2006*). Multiple motors carry cargos with increased processivity relative to single motors (*Derr et al., 2012*), which is essential for long-range transport in cell types such as neurons. Teams of motors also exert higher forces, which may enable the transport of large cargos through the dense cellular environment (*Blehm et al., 2013*). It has been proposed that cargos with multiple motors also avoid obstacles more effectively than single motors (*Hancock, 2014*). A recent study found that multi-kinesin cargos pause at MT defects rather than detach like single motors (*Liang et al., 2016*). Ensembles of two kinesin-1 motors linked together with a DNA scaffold have a higher run length than single kinesins, but their run length was also decreased in the presence of neutravidin obstacles on the MT (*Feng et al., 2018*). Motility of multiple dyneins has not been studied on MTs decorated with obstacles. Therefore, it is not well understood whether cargos driven by multiple motors bypass obstacles more successfully than single motors.

Here, we challenge single- and multi-motor cargos of kinesin and dynein with quantum dot (QD) and tubulin antibody obstacles on MTs. We find that kinesin and dynein employ different mechanisms to bypass these obstacles. Consistent with their ability to take side-steps, single dynein motors efficiently navigate around obstacles. Unlike dynein, single kinesins are strongly inhibited by QDs and antibodies on MTs yet overcome this limitation when working together as a team. The multiplicity of motors was also critical to bypass large roadblocks. When cargos driven by a team of kinesin or dynein motors face a wall along their path, they swing around the MT without net forward movement and continue moving forward. Together, our results provide insight into how motors transport intracellular cargos along obstacle-coated MT surfaces.

## Results

### Single dyneins, but not kinesins, bypass obstacles

Previous studies used rigor motors (*Schneider et al., 2015*), cell extracts (*Telley et al., 2009*) or MAPs (*Dixit et al., 2008*) to study how motors move in the presence of obstacles. Because some of these obstacles have complex binding kinetics to MTs, vary in size, and interact with motors directly, it is difficult to discern how their presence on the

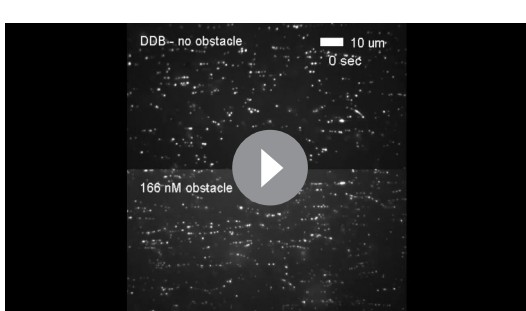

**Video 1.** Motility of DDB motors is not strongly affected by the QD obstacles. DDB motors were labeled with QD655 at their N-termini. Single-molecule motility of DDB in the presence of no obstacles (top) or 25 nM QD585 obstacles (bottom) on surface-immobilized MTs. The fluorescence signal of QD585 obstacles was collected in a separate channel, and not displayed. The sample was excited with a 1.7 kW cm$^{-2}$488 nm laser beam under the epifluorescence mode. Images were acquired at 10 Hz.
DOI: https://doi.org/10.7554/eLife.48629.004

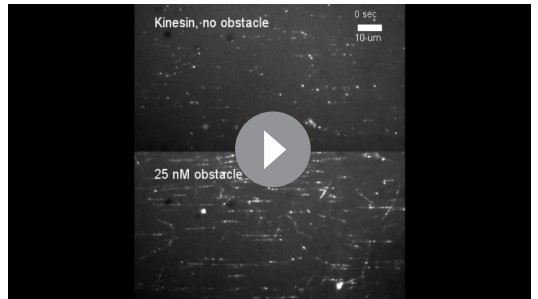

**Video 2.** Kinesin motors walk processively in the absence of QD obstacles, but motility was completely impaired in the presence of 25 nM QDs. Kinesin motors were labeled with QD655 at their C-termini. Single-molecule motility of kinesin in the presence of no obstacles (top) or 25 nM QD585 obstacles (bottom) on surface-immobilized MTs. The fluorescence signal of QD585 obstacles was collected in a separate channel, and not displayed. The sample was excited with a 1.7 kW cm$^{-2}$488 nm laser beam under the epifluorescence mode. Images were acquired at 10 Hz.
DOI: https://doi.org/10.7554/eLife.48629.005

MT obstructs motility. We sought a model obstacle that stably attached to the MT, had a well-defined size and formed no specific interactions with motors. To this end, we decorated biotinylated MTs with streptavidin-conjugated QDs (25 nm in diameter) (*DeWitt et al., 2012*). These QDs have a bright and photostable fluorescent emission, which enable us to measure their linear density along MTs.

We studied the motility of human kinesin-1, yeast cytoplasmic dynein, and mammalian dynein/dynactin/BicD2N (DDB) in the presence and absence of QD obstacles. Human kinesin-1 was truncated at its coiled-coil stalk (expressing amino acids 1–560) and fused to GFP and HaloTag at its C-terminus (*Belyy et al., 2016*). The N-terminal tail domain of yeast cytoplasmic dynein was replaced with a glutathione S-transferase (GST) tag for dimerization and a HaloTag for labeling (GST–Dyn331kD, expressing amino acids 1219–4093 of the dynein heavy chain) (*Reck-Peterson et al., 2006*). The DDB complex was assembled using full-length human cytoplasmic dynein, pig brain dynactin and N-terminal coiled-coil of mouse BicD2 (BicD2N). To facilitate DDB assembly, we used a human dynein mutant that does not form the autoinhibited conformation (*Zhang et al., 2017*). Motors were labeled with excess QDs of a different color on their tail region, and their motility on surface-immobilized MTs was monitored using multicolor imaging (*Figure 1A*). The run lengths of kinesin-QD complexes (2.1 ± 0.2 μm, mean ± SD) were similar to kinesin motors labeled with an organic dye (p=0.21, two-tailed t-test, *Figure 1—figure supplement 1D,E*), suggesting that QDs were driven by single motors.

The surface density of QD obstacles was varied, with a maximum decoration of 12 QDs $\mu m^{-1}$ (*Figure 1B,C*). Consistent with dynein's ability to take side-steps, we found that single yeast dynein and DDB motors walked processively even at the highest QD density tested (*Figure 1D,E*, *Figure 1—figure supplement 1*, *Video 1*) (*DeWitt et al., 2012*). We did not see evidence of motor reversals when DDB encountered a QD obstacle, suggesting that previously observed reversals of mammalian dynein might be due to diffusive motion of this motor in the absence of dynactin and a cargo adaptor (*McKenney et al., 2014*; *Schlager et al., 2014*; *Trokter et al., 2012*). While mobile fraction, velocity and run length of yeast dynein were reduced 30–70% by increasing density of QDs, DDB motility was less sensitive to obstacles. In comparison to dynein motors, kinesin motility was severely affected by the QDs (*Figure 1D,E*). The majority of kinesins became stuck on the MT with the addition of QDs (*Figure 1D,E*). At low QD density (1–2 $\mu m^{-1}$), the mobile fraction was reduced 90%, while the run length and velocity were reduced by 60% compared to the 0 QDs $\mu m^{-1}$ (p=0.0003, *Figure 1E*, *Video 2*). Kinesin motility could not be analyzed at higher QD densities because we did not detect processive runs longer than 250 nm under these conditions. Collectively, these results show that kinesin remains bound to an MT but is unable to move forward when it encounters a QD obstacle.

## Kinesin pauses longer than dynein when encountering QD obstacles

We next investigated how obstacles affected the pausing behavior of motors. Even at the lowest QD density, most kinesin motors were immotile throughout the recording, suggesting that kinesin has a high likelihood of permanently pausing when encountering a QD. Trajectories of the remaining processive motors were interspersed with frequent pauses (*Figure 2—figure supplement 1*). We analyzed the trajectories of these motors before they permanently paused or dissociated from the MT and calculated the residence times of motors per distance traveled (*Figure 2A*). Residence times were composed of two distinct states. A fast state corresponded to processive motility of the motor along MTs, and a slow state represented transient pauses in motility (*Figure 2B*). We calculated the density and length of pauses from the frequency and decay time of the slow state (*Figure 2B*). Strikingly, kinesin pause density increased two-fold and pause time increased four-fold at 2 QDs $\mu m^{-1}$. In contrast, pause density and duration of yeast dynein and DDB were only modestly increased by the QD density (*Figure 2C*). Transient pauses may correspond to detachment of kinesin when encountering an obstacle and reattachment to a nearby protofilament (*Schneider et al., 2015*). However, this mechanism is not robust enough to efficiently bypass obstacles, and kinesin motility stalls permanently usually after a few transient pauses in motility. Dynein also pauses frequently in the presence and absence of QD obstacles (*Figure 2—figure supplement 2*). However, unlike kinesin, it rarely pauses permanently even at the highest density of QDs (*Figure 1D,E*).

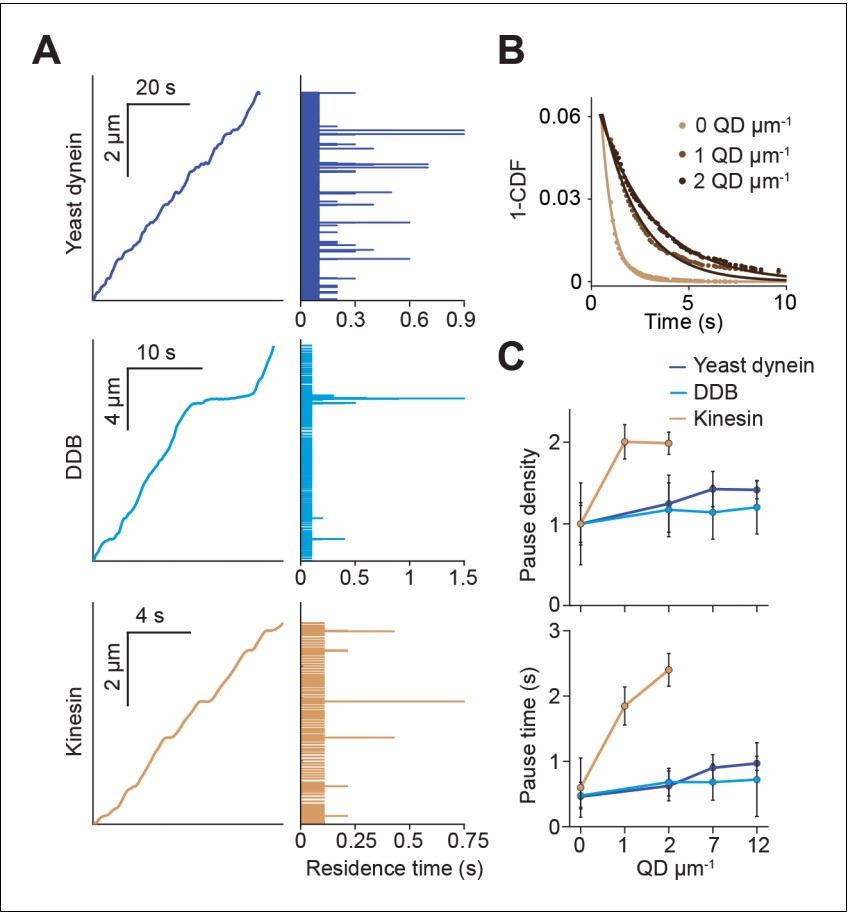

**Figure 2.** Kinesin pauses more frequently than dynein when encountering QD obstacles. (A) (Left) Representative traces of yeast dynein, DDB, and kinesin in the absence of QD obstacles on surface-immobilized MTs. (Right) Residence times of the motors in each section of the traces. (B) The inverse cumulative distribution (1-CDF) of kinesin residence times at different obstacle concentrations were fit to a single exponential decay. The residuals of that fit (shown here) are fit to a single exponential decay (solid line) to calculate the density and duration of kinesin pausing. (C) Density and duration of the pauses of the three motors. Pause densities (pauses/μm) are normalized to the 0 QDs μm$^{-1}$ condition. Kinesin pausing behavior at 7 and 12 QDs μm$^{-1}$ could not be determined because the motor was nearly immobile under these conditions. From left to right, $n$ = 535, 520, 158, 29 for yeast dynein, 511, 449, 391, 276 for DDB, and 570, 127, 112 for kinesin. Error bars represent SEM calculated from single exponential fit to residence times.

DOI: https://doi.org/10.7554/eLife.48629.006
The following figure supplements are available for figure 2:

**Figure supplement 1.** Kinesin pauses in the presence of QD obstacles.
DOI: https://doi.org/10.7554/eLife.48629.007
**Figure supplement 2.** Simulations for the pause analysis.
DOI: https://doi.org/10.7554/eLife.48629.008

## Kinesin quickly detaches from MTs decorated with antibody obstacles

Previous studies have reported that kinesin motors detach when encountering catalytically 'dead' motors or MAPs on the MT rather than getting stuck (*Telley et al., 2009*; *Schneider et al., 2015*). It is possible that bulky obstacles such as QDs may hinder motor movement more than proteins. To test this idea, we labeled kinesin motors with Cy3 dye instead of a QD. In assembling the dynein complex, mouse BicDR1 with a C-terminal SNAP tag was labeled with LD555 and incubated with dynein/dynactin (*Urnavicius et al., 2018*). In addition, we decorated the MTs with anti-tubulin antibody obstacles (*Figure 3A*). Because antibodies are small in size (~150 kDa), they likely block fewer protofilaments than QD obstacles (*Figure 3—figure supplement 1*).

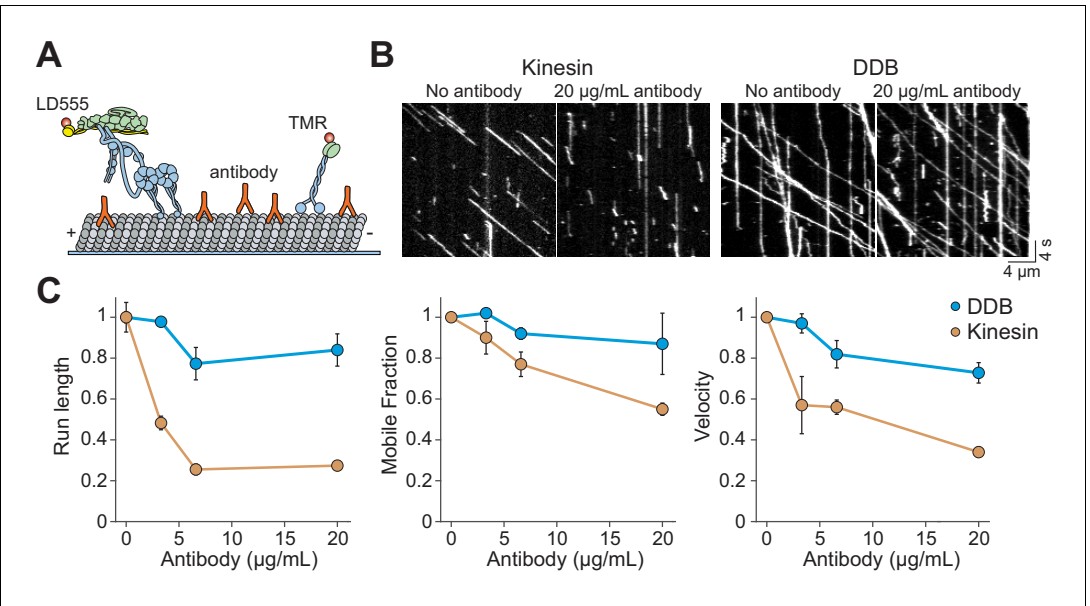

**Figure 3.** Kinesin detaches from MTs when encountering antibody obstacles. (**A**) Kinesin and dynein were labeled with organic dyes and their motility was tested in the presence and absence of anti-tubulin antibody on MTs. (**B**) Kymographs of TMR-kinesin and LD555-DDB walking on MTs in the absence and presence of 20 μg/mL antibody obstacle. (**C**) Quantification of how antibody obstacles affect motor motility. All data are normalized to the no antibody condition (mean ± SD, two independent experiments). From left to right, *n* = 185, 232, 199, 197 motors for kinesin and 104, 224, 262, 308 motors for DDB.

DOI: https://doi.org/10.7554/eLife.48629.009

The following figure supplement is available for figure 3:

**Figure supplement 1.** Larger obstacles block access to more protofilaments on an MT.
DOI: https://doi.org/10.7554/eLife.48629.010

Similar to QD obstacles, we see that single kinesin motors are more strongly inhibited by antibody obstacles than DDB (p=0.013 for velocity and 0.01 for run length, two-tailed t-test, *Figure 3B,C*, *Videos 3–4*). Kinesin run length and velocity were reduced by ~70% at 20 μg/mL antibody (p=0.01 and 0.0001, respectively,

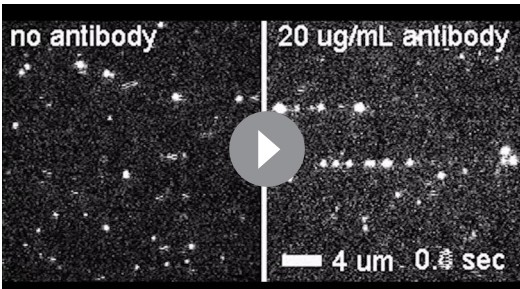

**Video 3.** Kinesin motors labeled with an organic dye are inhibited by antibody obstacles. Kinesin motors labeled with TMR walk on MTs in the presence (right) and absence (left) of antibody obstacles. MT fluorescence was recorded in a separate channel (not shown). There was a notable reduction in velocity and run length in the presence of antibody on the MT. Images were acquired at 5 Hz under TIRF illumination.
DOI: https://doi.org/10.7554/eLife.48629.011

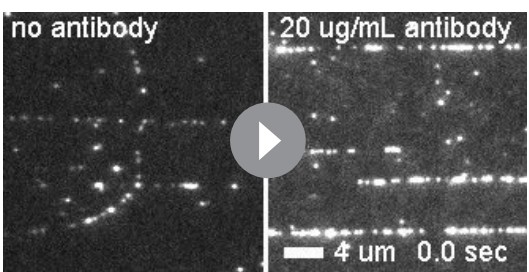

**Video 4.** DDB motors labeled with an organic dye are not strongly inhibited by antibody obstacles. DDB motors labeled with LD555 walk on MTs in the presence (right) and absence (left) of tubulin antibody obstacles. MT fluorescence was recorded in a separate channel (not shown). DDB motors were less affected by antibody obstacles than kinesin. Images were acquired at 5 Hz under TIRF illumination.
DOI: https://doi.org/10.7554/eLife.48629.012

two-tailed t-test, *Figure 3C*). In contrast to QD-obstacles, there was not a significant difference between the mobile fraction of kinesin and dynein at 20 µg/mL antibody (p=0.15, two-tailed t-test). Consistent with a previous study that used rigor kinesin as an obstacle (*Schneider et al., 2015*), we did not observe extended pauses of kinesin motors. Thus, antibody obstacles inhibit kinesin by causing them to detach rather than pausing for extended periods. Collectively, our results show that single kinesins detach from MT when encountering small protein obstacles. However, kinesin motors that carry a rigid QD cargo are more likely to pause when encountering a bulkier obstacle on an MT.

## Obstacle avoidance of single motors on freely suspended MTs

On surface-immobilized MTs, motors cannot access protofilaments facing the coverslip. As a result, surface immobilization may serve as an additional obstacle as the motors attempt to bypass the QDs. MTs in the cell, however, are freely suspended in 3D. This may allow motors to fully explore the MT surface and more successfully bypass the obstacles. To test this possibility, we constructed 'MT bridges' by immobilizing MT ends to polydimethylsiloxane (PDMS) ridges on either end of a 10 µm deep valley (*Figure 4A,B*, *Figure 4—figure supplement 1*). Similar to surface-immobilized MTs, DDB and yeast dynein were able to walk at the highest QD concentration tested on MT bridges (*Figure 4C*). Interestingly, yeast dynein's run length was 2.5-fold higher on MT bridges compared to surface-immobilized MTs at 12 QD $\mu m^{-1}$ (p<0.001, two-tailed t-test, *Figure 1E*, *Figure 4C*), suggesting that this motor bypasses obstacles more successfully by exploring the entire MT surface. However, we did not observe a significant improvement in kinesin motility on MT bridges in comparison to surface-immobilized MTs (*Figure 1E*, *Figure 4C*). The mobile fraction was reduced by 75% at 1 QD $\mu m^{-1}$ compared to the no obstacle condition, and motility could not be detected at 7 QDs $\mu m^{-1}$. Similar to surface-immobilized MTs, frequent pauses were observed in kinesin motility in the presence of QD obstacles on MT bridges (*Figure 4—figure supplement 2*). These results suggest that kinesin is intrinsically limited by its ability to side-step to adjacent protofilaments when it encounters obstacles on an MT.

## Multi-kinesin cargos bypass obstacles on MTs

In cells, cargos are often carried by multiple motors, which increases collective force generation and enables transport of the cargo over longer distances (*Derr et al., 2012*; *Rai et al., 2016*; *Encalada et al., 2011*; *Levi et al., 2006*) as well as at slightly higher velocities (*Nelson et al., 2014*). We asked whether multiple motors can transport cargo under conditions in which single motors are unable to walk along MTs. To test this, 500 nm cargo beads were coated with multiple kinesins or DDB motors (*Figure 5A*). In the absence of QD obstacles, the beads were highly processive and did not detach until they reached the end of the MT. When the beads were incubated with a low concentration (50 nM) of kinesin motors, we detected processive motility of beads, albeit with frequent pausing, on MTs decorated with 7 QDs $\mu m^{-1}$ (*Figure 5B*). This was a density at which single kinesins were completely inhibited (*Figure 1E*). However, motility of these beads was severely inhibited at 12 QDs $\mu m^{-1}$. Surprisingly, when beads were incubated with a higher concentration (1.5 µM) of kinesin, their mobile fraction was unaffected by the decoration of MTs with 12 QDs $\mu m^{-1}$ (*Figure 5B,C*, *Figure 5—figure supplement 1*, *Video 5*). Similarly, multiple DDBs transported beads to the minus-ends of MTs regardless of the surface density of QDs with no decrease in mobile fraction (*Figure 5B,C*).

The analysis of the individual trajectories of dynein-driven beads showed that the velocity decreased by 28% in the presence 12 QD $\mu m^{-1}$ (p=0.02, two-tailed t-test, *Figure 5C*), comparable to a 19% decrease of the velocity of single dyneins under the same conditions (p=0.05, two-tailed t-test, *Figure 1E*). The velocity of kinesin-driven beads also decreased by 25% decrease when exposed to 12 QDs $\mu m^{-1}$ (p=0.02, two-tailed t-test, *Figure 5C*). We also tested if multi-kinesin cargoes were able to walk on antibody-coated MTs better than single kinesin motors. For beads coated with 1.5 µM of kinesin, run length and mobile fraction were unaffected by 20 µg/mL antibody (*Figure 5D,E*). Remarkably, all beads walked until they reached the end of the MT (*Figure 5D*, *Video 6*). Mobile fraction was also not reduced (p=0.73, *Figure 5E*). Similar to QD-obstacles, multi-motor kinesin exhibited reduced velocity at 20 µg/mL antibody (p=0.04, *Figure 5E*). Therefore, while

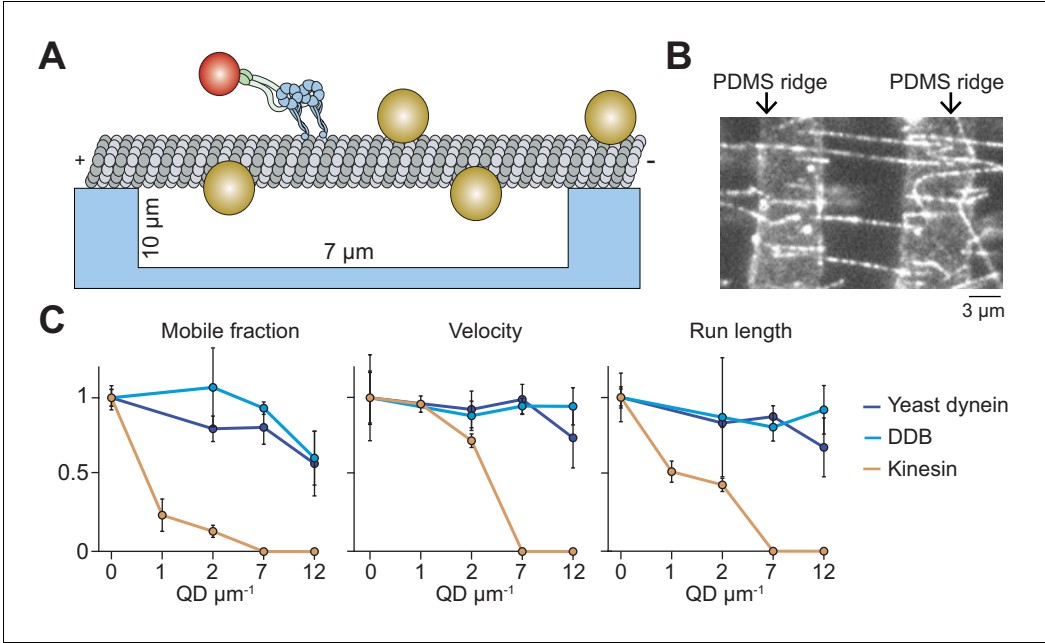

**Figure 4.** Suspending MTs from the surface does not aid kinesin in avoiding obstacles. (**A**) Schematic of a single-molecule motility assay on MT bridges coated with QD obstacles (not to scale). (**B**) An example image of Cy5-labeled MT bridges in the microfabricated chamber. PDMS ridges (arrows) are visible due to the autofluorescence. (**C**) Mobile fraction, velocity and run length of motors along MT bridges were normalized to the no QD condition (mean ± SD, three independent experiments). From left to right, *n* = 199, 187, 106 for kinesin, 129, 107, 163, 135 for yeast dynein, and 192, 206, 330, 276 for DDB.

DOI: https://doi.org/10.7554/eLife.48629.013

The following figure supplements are available for figure 4:

**Figure supplement 1.** Chamber design and raw data of single-molecule motility along MT bridges.

DOI: https://doi.org/10.7554/eLife.48629.014

**Figure supplement 2.** Kinesin pauses in the presence of QD obstacles on suspended MTs.

DOI: https://doi.org/10.7554/eLife.48629.015

single kinesins are strongly affected by obstacles on MTs, a team of kinesins can carry cargo beads over long distances along MTs densely decorated with obstacles as well as dyneins.

## Multi-motor cargos rotate around the MT to avoid large obstacles

Avoiding obstacles larger than a QD, such as a stationary organelle or intersecting cytoskeletal filament, may require cargoes to rotate to the other side of the MT before continuing forward movement (*Verdeny-Vilanova et al., 2017*). In vivo studies have observed both anterograde and retrograde cargos to bypass stationary organelles (*Che et al., 2016*). It has been proposed that rotation of cargoes around the MT requires the presence of both kinesin and dynein motors on the cargo or the distortion of the lipid cargo (*Hancock, 2014*; *Kaplan et al., 2018*; *Verdeny-Vilanova et al., 2017*). To test whether a single type of motor can rotate a rigid cargo around the MT, we tracked beads driven by multiple kinesins or dyneins on MT bridges. If the beads were positioned below the MT when they reached the end of the bridge, they were challenged to bypass the PDMS wall (*Figure 6A*). Remarkably, we observed that most of these beads rotated to the top of the MT with no forward motion before they continued along the MTs (77 ± 13% kinesin beads and 85 ± 5% dynein beads, mean ± SD, *Figure 6B,C*, *Videos 7* and *8*). This movement was different from the previously observed helical movement of kinesin- or dynein-driven cargos around the MT (*Can et al., 2014*; *Nitzsche et al., 2008*), in which rotation is accompanied by forward translational movement. 29% and 24% of kinesin- and dynein-driven beads, respectively, paused before moving forward (*Figure 6B,C*), similar to intracellular cargos that encounter stationary organelles (*Che et al.,*

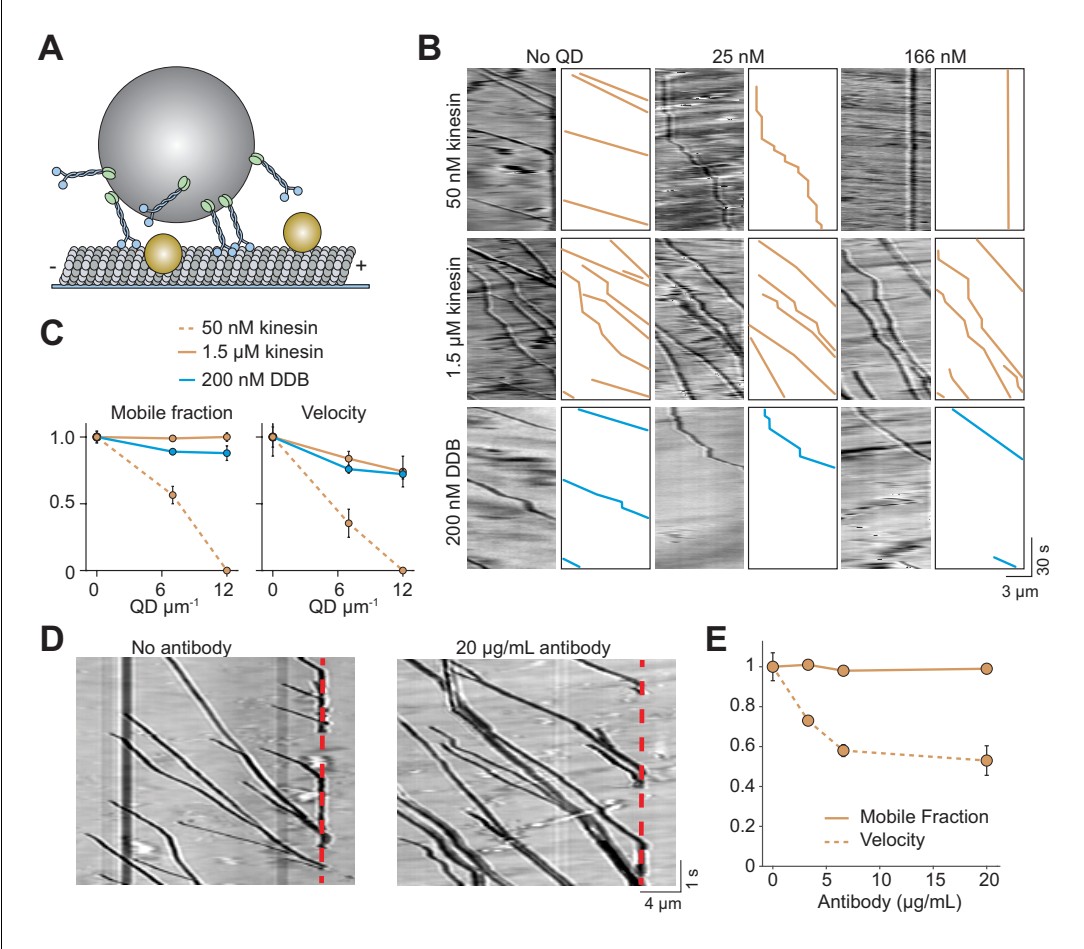

**Figure 5.** Cargos driven by multiple kinesins successfully bypass obstacles. (**A**) Schematic of bead motility driven by multiple motors along surface-immobilized MTs decorated with QD obstacles (not to scale). (**B**) Kymographs reveal the motility of beads coated with kinesin or DDB in the presence and absence of QD obstacles. Diffusion of unattached beads creates a background in the kymograph. To the right of each kymograph is an illustration of the bead motility on the MT. (**C**) Mobile fraction and velocity of beads were normalized to the no QD condition (mean ± SD). From left to right, n = 154, 189 bead traces for 50 nM kinesin, 323, 338, 336 for 1.5 μM kinesin, and 279, 184, 67 for DDB from three independent experiments. (**D**) Representative traces for 1.5 μM kinesin on beads in the absence and presence of antibody obstacle. The dashed red line indicates the plus-end of the MT. (**E**) Quantification of mobile fraction and velocity of kinesin-driven beads. From left to right, n = 145, 198, 141, 201 beads.
DOI: https://doi.org/10.7554/eLife.48629.016

The following figure supplement is available for figure 5:

**Figure supplement 1.** The analysis of beads driven by multiple motors on surface-immobilized MTs without normalization.
DOI: https://doi.org/10.7554/eLife.48629.017

*2016*). In contrast to strong inhibition of single kinesins by obstacles, beads driven by multiple kinesins paused for a shorter period than dynein-driven beads when they encountered the wall (3.8 ± 0.2 s vs 7.8 ± 1.0 s, mean ± SEM, *Figure 6D*). Only 23% of kinesin beads and 15% of dynein beads either got stuck or detached from MTs at the PDMS wall. We concluded that cargos driven by multiple motors can bypass large obstacles by rotating around the circumference of the MT.

## Discussion

Despite the complexity of the cytoplasm, kinesin and dynein drive intracellular transport with remarkable efficiency towards MT ends. In this study, we investigated the ability of kinesin and dynein to bypass permanent obstacles using in vitro reconstitution and single-molecule imaging.

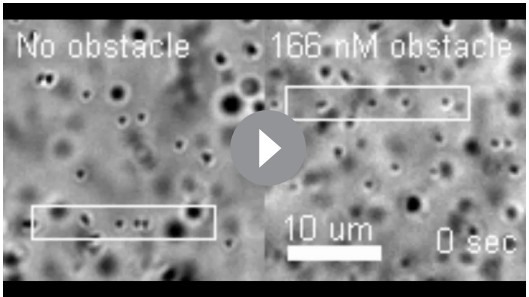
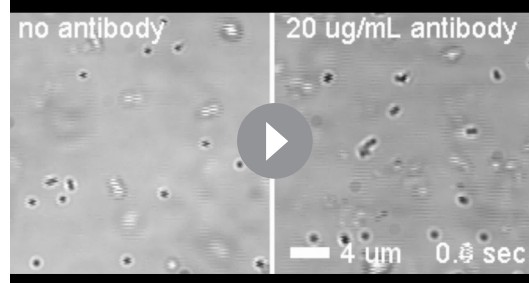

**Video 5.** Beads driven by multiple kinesins move processively at high QD obstacle concentrations. 500 nm diameter beads were labeled with 1.5 µM kinesin. Beads move along surface-immobilized MTs in the presence of no obstacles (left) or 166 nM obstacles (right). Boxes highlight the processive motility of beads along a surface-immobilized MTs (not labeled). In addition, freely diffusing beads in the chamber come in and out of focus during imaging. Images were acquired at 10 Hz under brightfield illumination.
DOI: https://doi.org/10.7554/eLife.48629.018

**Video 6.** Beads driven by multiple kinesin motors walk processively on MTs coated with antibody obstacle. Beads coated with 1.5 µM kinesin motors walk on MTs in the presence (right) and absence (left) of antibody obstacles. MT fluorescence was recorded in a separate channel (not shown). Images were acquired at 5 Hz under brightfield illumination. Motility of monodisperse beads was analyzed, while clumps of beads were excluded from the analysis.
DOI: https://doi.org/10.7554/eLife.48629.019

These results show that kinesin and dynein utilize different mechanisms to bypass these obstacles. Despite their large size, single dyneins were highly capable of maneuvering around obstacles on the MT. In contrast, single kinesins were strongly inhibited by obstacles, as previously reported (*Dixit et al., 2008*; *Leduc et al., 2012*; *Telley et al., 2009*; *Schneider et al., 2015*). These results are likely a consequence of differences in the sidestepping ability of the two motors (*Can et al., 2014*; *DeWitt et al., 2012*; *Ray et al., 1993*). Remarkably, multiple kinesins were able to bypass these obstacles as efficiently as dyneins. In the case of multi-motor cargos, we anticipate that kinesin motors are just as likely to get stuck at an obstacle as a single motor. However, the other motors driving the cargo may exert a force on the stuck motor, causing its rapid detachment and the continued forward motion of the cargo (*Tjioe et al., 2019*). In this process, new motors in the leading direction are recruited to the lattice while the motors in the trailing direction detach from the MT. Alternatively, a team of motors carrying the cargo pause until they move sideways either by stepping to neighboring protofilaments or because stochastic shifts in the center of mass of the bead allow for unbound motors to attach to adjacent protofilaments. Addressing the mechanism of obstacle avoidance of multiple motors will require future studies at higher spatial resolution.

Multi-motor teamwork also proved beneficial for both types of motors when challenged by a large obstacle. Cargo beads driven by multiple kinesin or dynein motors were able to maneuver around a PDMS obstruction that blocked access to half of the MT surface (*Figure 3—figure supplement 1*). This behavior helps explain how cargoes bypass large cellular roadblocks such as stationary organelles or intersecting cytoskeletal filaments. Recent measurements observed rotational motion in the trajectories of endosomal cargos carrying gold nanorods (*Kaplan et al., 2018*). In addition, correlative live-cell and super-resolution microscopy showed that rotational movement could be used by cargos to avoid steric obstacles (*Verdeny-Vilanova et al., 2017*). These studies proposed that rotational movement and off-axis stepping might result from having a mix of motors, such as kinesin-2 or dynein, on the cargo along with kinesin-1. While transient back-and-forth movement of cargo may allow it to change the protofilament track, our results clearly show that tug-of-war between opposite polarity motors is not required to bypass large obstacles. Instead, cargo beads driven by multiple motors can switch to the other side of the MT surface by rotation, when only one type of motor is active at a time. We also showed that fluidity of the cargo is not essential for this process (*Verdeny-Vilanova et al., 2017*). These results show that multiplicity of motors not only increases the collective force generation and the length of processive runs on an MT but also enables motors to maneuver around obstacles in their path. Future studies are required to address how the motor copy number affects the ability of cargoes to bypass dynamic obstacles, such as MAPs.

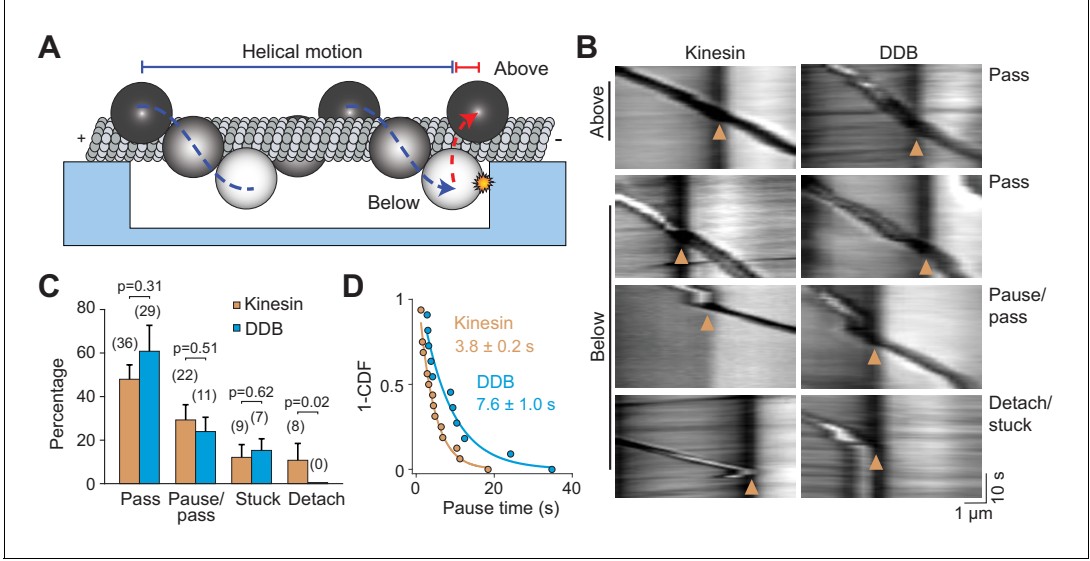

**Figure 6.** Cargos driven by multiple motors bypass large obstacles by rotating around the MT. (A) Schematic of multi-motor bead motility on MT bridges. The position of the bead in the *z*-axis is determined from changes in bead intensity under brightfield illumination. If a bead is positioned below the MT when it reaches the PDMS wall, it must move to the top of the MT (red dotted curve) before continuing forward. (B) Kymographs reveal how beads driven by kinesins and DDBs move when they encounter the PDMS wall (below). (Pass) The bead rotates around the MT as evidenced by light to dark transitions in the bead intensity at the wall before continuing forward. (Pause/pass) The bead paused for more than 1 s at the wall before rotating around the MT and moving forward. (Detach/stuck) The bead failed to pass the wall and either detached (left) or got stuck on an MT (right). (C) The percentage of the pass, pause/pass and detach/stuck events for the beads positioned below the MT when they encounter the PDMS wall (mean ± SD, two independent experiments). The number of beads is shown in parentheses. p-values are calculated using two-tailed t-test for pass, pause/pass and stuck, and z-score calculation for detach. (D) The inverse cumulative distribution of pause times for kinesin and DDB beads. A fit to a single exponential decay (solid curves) revealed that pause duration of DDB-driven beads is longer than kinesin-driven beads (F-test, p=0.0001, *n* = 16 pauses for kinesin and 11 for DDB).

DOI: https://doi.org/10.7554/eLife.48629.020

# Materials and methods

## Key resources table

| Reagent type (species) or resource | Designation | Source or reference | Identifiers | Additional information |
| --- | --- | --- | --- | --- |
| Other | Amino quantum dot (655) | ThermoFisher | Q21521MP | |
| Other | Streptavidin quantum dot (585) | ThermoFisher | Q10111MP | |
| Chemical | APTES | Sigma | 440140 | |
| Antibody | Anti-tubulin antibody (mouse monoclonal) | Sigma, Tub 2.1 | T5201 | Dilution range 0–20 µg/mL |
| Peptide, recombinant protein | Human Kinesin-1 | *Belyy et al., 2016* | N/A | |
| Peptide, recombinant protein | Yeast dynein heavy chain | *Reck-Peterson et al., 2006* | N/A | |
| Peptide, recombinant protein | BicD2 (amino acids 1–400) | *Schlager et al., 2014* | Addgene 111862 | |

*Continued on next page*

*Continued*

| Reagent type (species) or resource | Designation | Source or reference | Identifiers | Additional information |
|---|---|---|---|---|
| Peptide, recombinant protein | Human cytoplasmic dynein complex | *Zhang et al., 2017* | N/A | |
| Chemical | Acetone | Sigma | 270725 | |
| Chemical | Ethanol | Sigma | 459828 | |
| Other | IgG Sepharose Beads | GE Healthcare | 17096902 | |
| Chemical | Glutaraldehyde | Fisher Scientific | G1511 | |
| Other | PDMS | Sylgard 184 Silicone Elastomer | N/A | |
| Other | Glucose oxidase | Sigma | G2133 | |
| Other | Catalase | Sigma | C3155 | |
| Chemical | Taxol | Sigma | T7191 | |
| Peptide, recombinant protein | Pig Brain Dynactin | *Schlager et al., 2014* | N/A | |
| Peptide, recombinant protein | Pig Brain Tubulin | *Castoldi and Popov, 2003* | N/A | |
| Chemical | ATP | Sigma | A3377 | |
| Other | Ni-NTA beads | Thermo Scientific | 88221 | |
| Chemical | Fugene HD transfection reagent | Promega | E2311 | |
| Chemical | HaloTag Ligand succinimidyl ester | Promega | P6751 | |
| Chemical | SU-8 2010 photoresist | Microchem | N/A | |
| Other | Super Active Latex Beads | Thermo Fisher | C37481 | |
| Chemical | Sulfo-NHS | Thermo Fisher | 24510 | |
| Chemical | EDC | Thermo Fisher | 22980 | |
| Software | U-track | *Jaqaman et al., 2008* | N/A | |
| Antibody | Anti-GFP (anti-rabbit polyclonal) | Covance | N/A | Used at 0.4 mg/mL |
| Peptide, recombinant protein | Mouse BicDR1 (full length) | *Urnavicius et al., 2018* | Adapted from Addgene 111585 | |

## Protein expression and purification

A human kinesin-1 coding sequence expressing amino acids 1–560 was fused to GFP, HaloTag and a 6xHis tag on the C-terminus (hK560::GFP::HaloTag::6xHis) (*Belyy et al., 2016*). The N-terminus of yeast cytoplasmic dynein was replaced with a HaloTag and a GST dimerization tag (HaloTag-GST-Dyn1$_{331kDa}$, consisted of amino acids 1219–4093 of the dynein heavy chain) (*Reck-Peterson et al., 2006*). A full length human cytoplasmic dynein construct consisted of the dynein heavy chain tagged with an N-terminal SnapTag, cloned into a pOmniBac vector, and fused to a plasmid that contained dynein intermediate chain, light intermediate chain and three different light chains (Tctex, Roadblock and LC8), as described (*Schlager et al., 2014*). We used a dynein mutant that does not form the autoinhibited phi-conformation (*Zhang et al., 2017*) to facilitate assembly of the DDB complex. The

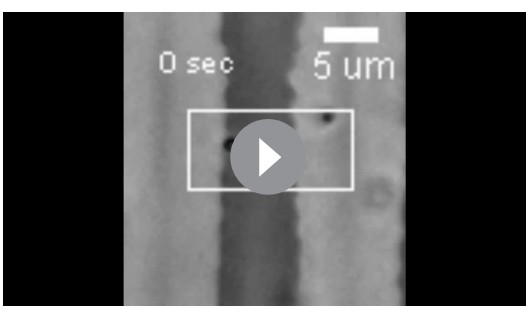 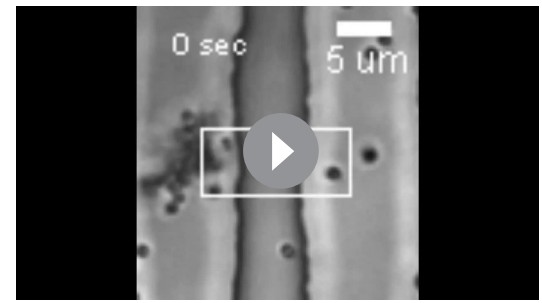

**Video 7.** Beads driven by multiple kinesins bypass the PDMS wall. The box highlights the processive motility of a bead driven by multiple kinesins on an MT bridge (unlabeled) suspended over the PDMS ridges. The valley (dark) is in the center of the movie while the PDMS ridges (light) are on either side. As the bead reaches the PDMS wall (light/dark interface), the bead intensity shifts from light to dark before the bead continues to walk on the PDMS ridge. Images were acquired at 10 Hz under brightfield illumination.
DOI: https://doi.org/10.7554/eLife.48629.021

**Video 8.** Beads driven by multiple DDBs bypass the PDMS wall. The box highlights the processive motility of a bead driven by multiple DDBs on an MT bridge (unlabeled) suspended over the PDMS ridges. The valley (dark) is in the center of the movie while the PDMS ridges (light) are on either side. As the bead reaches the PDMS wall (light/dark interface), the bead intensity shifts from light to dark before the bead continues to walk on the PDMS ridge. Images were acquired at 10 Hz under brightfield illumination.
DOI: https://doi.org/10.7554/eLife.48629.022

BicD2N construct consisted of GFP fused to the first 400 amino acids of mouse BicD2 (*Schlager et al., 2014*). The mouse BicDR1consisted of SNAP tag fused to the C-terminus of the full-length protein (*Urnavicius et al., 2018*).

## Kinesin purification

Rosetta cells transformed with kinesin plasmid were grown in a 5 mL culture overnight. This culture was added to 1 L of LB media and grown for 3 hr until the OD600 reached 0.7. Cells were induced with fresh 100 µM IPTG, put on ice until the temperature reached 20 degrees Celcius and incubated overnight at 20 degrees at 180 RPM. After harvesting cells at 4785 RCF for 15 min in a JLA 8.1 rotor, 500 mL cell pellets were incubated with 40 mL lysis buffer (50 mM sodium phosphate monobasic pH 8.0, 250 mM sodium chloride, 2 mM magnesium chloride, 20 mM imidazole, 1 mM ATP, 10 mM beta-mercaptoethanol (BME), 1 mM phenylmethylsulfonyl fluoride (PMSF)). Cells were lysed with a sonicator and spun in a Ti70 rotor at 117734 RCF for 30 min. The supernatant was incubated with 6 mL of washed Ni-NTA bead slurry (HisPur, Thermo Scientific) for 1 hr at four degrees. Beads were collected in a BioRad column and washed in wash buffer (50 mM sodium phosphate monobasic pH 6, 250 mM sodium chloride, 1 mM magnesium chloride, 20 mM imidazole, 100 µM ATP, 10 mM BME). Protein was eluted in elution buffer (50 mM sodium phosphate monobasic pH 7.2, 250 mM sodium chloride, 1 mM magnesium chloride, 500 mM imidazole, 100 µM ATP, 10 mM BME) and snap-frozen in liquid nitrogen after addition of 20% glycerol.

## Yeast dynein purification

Yeast cells were cultured on YPAD plates for 2–3 days. A 10 mL culture was grown overnight in YP media with 1 mL of 25% dextrose + 0.04% adenine supplements at 30 degrees. 2 mL of the culture was then added to 100 mL of 1.25x YP media supplemented with 10 mL of 20% raffinose. After 9 hr of growth, the entire culture was added to 1.8 L of YP media supplemented with 200 mL of 20% (w/v) galactose. Cells were cultured at 30 degrees with shaking (200 rpm) overnight until the OD600 reached 1.5. After harvesting cells at 4785 RCF for 15 min in a JLA 8.1 rotor, cells were frozen drop-wise and lysed while frozen in a coffee grinder. 50 mL of lysis buffer (30 mM HEPES pH 7.4, 50 mM potassium acetate, 2 mM magnesium acetate, 1 mM EGTA, 10% glycerol, 1 mM dithiothreitol, 100 µM ATP, 1 mM PMSF) was added to a 1 L yeast pellet. Cells were spun at 360562 RCF for 45 min in a Ti70 rotor. The supernatant was incubated with washed IgG beads (IgG Sepharose 6 Fast Flow, GE Healthcare) for 1 hr with gentle rolling. Beads were collected using a BioRad disposable column, washed with wash buffer (lysis buffer with 125 mM KCl) and TEV buffer (10 mM Tris pH 8, 150 mM

KCl, 10% glycerol, 1 mM tris(2-carboxyethyl)phosphine, 100 µM ATP, 1 mM PMSF). Beads were transferred to an Eppendorf tube and eluted with TEV protease for 1 hr. Beads were then spun down and the supernatant was snap-frozen in 20% glycerol.

## Purification of mammalian dynein-dynactin

Human dynein, mouse BicD2N, mouse BicDR1 and pig brain dynactin were purified as previously described (Zhang et al., 2017). Further information can also be found on Invitrogen's Bac-to-Bac Baculovirus Expression System Guide (Invitrogen). Briefly, plasmids containing genes of interest were transformed into DH10Bac competent cells and plated on LB agar plates with kanamycin, gentamycin, tetracycline, Blue-gal, and isopropyl beta-D-1-thiogalactopyranoside. An overnight culture of a colony grown in 2X YT media with kanamycin, gentamycin and tetracycline and the bacmid was purified from these cells. Cells were lysed and neutralized using Qiagen miniprep buffers P1, P2, and P3. DNA was then precipitated with isopropanol and spun down for 10 min at 13,000 RCF at 4°C. The DNA pellet was washed three times with 70% ethanol, air-dried and resuspended in Qiagen's EB buffer.

Bacmid was used within a few days for transfecting Sf9 cells. All insect cell culture was courtesy of Berkeley's Cell Culture Facility. The cells have not been authenticated or tested for mycoplasma contamination. 2 mL of Sf9 cells at 500,000 cells/mL was aliquoted into a 6-well dish and allowed to attach for 10 min. 1 microgram of bacmid DNA was diluted in ESF 921 media (Expression systems, no antibiotic or serum), mixed with 6 µL of Fugene HD transfection reagent (Promega) and incubated for 15 min at room temperature. Media on the cells was removed and replaced with 0.8 mL of ESF 921 media. The Fugene/DNA mix was added dropwise on the cells. The dish was sealed with Parafilm and incubated for 72 hr. 24 hr into this incubation, 1 mL of extra ESF 921 media was added to the cells. After removing the media and spinning, 1 mL of the supernatant (P1 virus) was added to 50 mL of Sf9 cells at a density of 1 million cells/mL. Following a 72 hr incubation, the media was spun down and the supernatant (P2 virus) was harvested. 10 mL of the P2 virus was used to infect 1 L of Sf9 cells at 1 million cells/mL and expression proceeded for 72 hr. Cells expressing the protein of interest were harvested at 4000 RCF for 10 min and resuspended in 50 mL lysis buffer (50 mM HEPES pH 7.4, 100 mM NaCl, 10% glycerol, 1 mM DTT, 100 µM ATP, 2 mM PMSF and 1 tablet of protease inhibitor cocktail). Lysis was performed using 15 loose and 15 tight plunges of a Wheaton glass dounce. The lysate was clarified using a 45 min, 360562 RCF spin in a Ti70 rotor and incubated with 2 mL IgG beads (IgG Sepharose 6 Fast Flow, GE Healthcare) for 2 hr. Beads were washed with lysis buffer and TEV buffer (50 mM Tris pH 7.4, 150 mM potassium acetate, 2 mM magnesium acetate, 1 mM EGTA, 10% glycerol, 1 mM dithiothreitol, 100 µM ATP). Beads were then collected and incubated with TEV protease overnight to elute the protein. Finally, the protein was concentrated and snap-frozen in liquid nitrogen after the addition of 20% glycerol.

## Glass silanization

Glass slides were functionalized with aminopropyltriethoxysilane (APTES) and glutaraldehyde to allow for covalent attachment of MTs, as described previously (Nicholas et al., 2014). APTES (Sigma, 440140) aliquots were prepared in 5 mL cryotubes (Corning, 430656) using glass pipettes, capped under nitrogen atmosphere, snap-frozen upright in liquid nitrogen and stored in −80°C. Glass slides were sonicated in a 2% Mucasol (Sigma, Z637181) prepared in hot water and then rinsed thoroughly in water. Slides were then baked on a hot plate (Benchmark, BSH1002) to remove excess water for 5 min. To create functional silanol groups, slides were treated with oxygen plasma (PETS Reactive Ion Etcher) at 200 mTorr oxygen, 55 W for 1 min. Slides were rinsed briefly in acetone (Sigma, 270725) and immersed in a 2% (v/v) APTES in acetone for 1 min. APTES aliquots were added to acetone before warming to room temperature. After silane treatment, the slides were rinsed in acetone and baked on a 110°C hot plate for 30 min. To remove silane unbound to the glass, slides were sonicated sequentially in ethanol (Sigma, 459828) and water for 5 min. Following this treatment, slides were again baked at 110°C for 30 min. An 8% glutaraldehyde solution (Fisher Chemical, G1511) was prepared in water and 1 mL drops of the solution were made on Parafilm. Slides were incubated functionalized-side down on the glutaraldehyde solution drops for 30 min in a sealed container. Finally, the slides were washed and sonicated in water for 10 s to remove loosely absorbed glutaraldehyde and stored in a sealed container at room temperature up to 1 week.

## Labeling

Motors were labeled with QDs modified with a HaloTag or SnapTag ligand. Amino-PEG-QDs (Thermo, Q21521MP) were labeled with HaloTag ligand by reaction with N-hydroxysulfosuccinimide reactive Halo-Tag ligand (Promega, P6751) or Snap-Tag ligand for 30 min at room temperature. QDs were then exchanged into 25 mM borate pH 8.5 using Amicon 100K centrifugal filters and stored in that buffer. 2 µM of these QDs were mixed with 100–500 nM motors fused with a SNAP-Tag or HaloTag for 10 min on ice. For labeling with dye, HaloTag-Cy3 was added to kinesin while it was bound to beads during the purification. Excess dye was washed away before eluting the protein. A similar procedure was used to label SNAP-BicDR1.

## Motility assays

Tubulin was purified from pig brain and labeled with biotin or fluorophores as described (*Castoldi and Popov, 2003*; *Nicholas et al., 2014*). The final percentage of biotin on the MT was less than 5%. To perform motility assays, biotinylated MTs were diluted in BRB80 (80 mM PIPES pH 6.8, 1 mM EGTA, 1 mM magnesium chloride) supplemented with 10 µM taxol and flowed into a chamber made with two pieces of double-sided tape between an APTES-silanized slide and an unmodified coverslip. Chamber was then passivated with BRB80 supplemented with 1 mg/mL casein (Sigma, C5890), 1 mM DTT and 10 µM taxol. The chamber was incubated with different dilutions of streptavidin-coated QDs (Invitrogen, Q10111MP) or monoclonal anti-β-tubulin antibody produced in mouse (T5201, Sigma). At low QD concentrations, the linear density of QD-obstacles on the MT was measured by counting the number of fluorescent spots from TIRF images. At higher QD concentrations, the linear density was estimated from the ratio of total fluorescence signal on an MT to the fluorescence intensity of a single QD.

BRB80 (above), DLB (30 mM HEPES pH 7.2, 2 mM magnesium chloride, 1 mM EDTA, 10% glycerol) and MB (30 mM HEPES pH 7.0, 5 mM MgSO$_4$, 1 mM EGTA) buffers were used for assaying the motility of kinesin, yeast dynein, and DDB, respectively. Motor-QD mixtures were flowed into the chamber, bound to the MT and washed to remove unbound motor and QD. For experiments with antibody obstacles, dye- labeled kinesin-1 or BicDR1 were used. Finally, motor-specific buffer supplemented with 1 mg/mL casein, 1 mM tris(2-carboxyethyl) phosphine (TCEP), 100 µM ATP, glucose oxidase, catalase, and 0.8% dextrose were flown into the chamber. Run-length and velocity were determined by selecting the beginning and end of each trace in ImageJ. Cumulative frequencies of run lengths were fit to a single exponential function. Reported run lengths are the half-life values. The mobile fraction was calculated by dividing the number of moving motors over the total number of motors observed in the kymograph.

## Microscopy

Microscopy was performed using a custom-built fluorescence microscope equipped with a Nikon Ti-E Eclipse microscope body, a 40 × 1.15 NA long-working-distance water immersion objective (Nikon, N40XLWD-NIR), and a perfect focusing system (*DeWitt et al., 2012*). The sample position was controlled using an automated microscope stage (Microstage 20E, MadCityLabs). The sample was excited in the epifluorescence mode using 488, 561 and 633 nm laser beams (Coherent). Fluorescence image was split into two channels using OptoSplit2 (Cairn instruments) and detected by an electron-multiplied CCD Camera (Andor Ixon, 512 × 512 pixels).

## Single-molecule tracking and pause analysis

Single-particle tracking was performed using Utrack (*Jaqaman et al., 2008*). Tracks were split into 1D motion along the long axis of the MT and perpendicular direction as described (*DeWitt et al., 2015*). All tracks were manually reviewed to exclude tracks with jumps greater than 100 nm. Localization error was calculated using high pass filtering of the trajectories by calculating $x_i' = \frac{x_{i+1} - x_i}{\sqrt{2}}$, where $x$ is the position of the probe along the MT axis and $i$ is the frame number. This operation omits unidirectional motility and pauses at lower frequencies, leaving only the Gaussian noise associated with the trace. $\sigma_{x'}$ is defined as the localization error. Under our imaging conditions, the localization error of QDs was typically between 20-40 nm (*Figure 2—figure supplement 1*). Pause analysis was performed as described (*DeWitt et al., 2015*). Briefly, the track was divided into distance bins and the residence time within each bin was calculated. All non-zero residence times were

fit to a single exponential decay. The residuals were then fit to a single exponential decay. The decay time and amplitude of this second fit were defined as the average pause time and pause density, respectively.

## Simulations

Optimum running window averaging and bin size were calculated from simulated traces generated in MatLab. Experimentally determined noise, velocity and pause distributions were used to generate traces (*Figure 2—figure supplement 1*). A particle takes 8 nm unidirectional steps with exponentially distributed dwell times of 0.015 and 0.59 s, mimicking the characteristic times of processive motility and pausing of kinesin motors, respectively. The trace was then resampled to the imaging rate of 10 Hz. Random Gaussian noise was added to each position to introduce localization error. Simulations were used to determine whether the localization error in traces interferes with pause detection.

## Microfabrication

PDMS bridges were generated using soft lithography (*Qin et al., 2010*; *Théry and Piel, 2009*). Briefly, SU-8 2010 negative photoresist (Microchem) was spun on to a silicon wafer to 10 µm thickness. After a soft bake, the photoresist was exposed to UV light through a patterned, film photomask (Fine Line Imaging) on an OAI 200 Lithographic Mask Aligner. The pattern was developed after a hard bake using SU-8 developer (Microchem). To render the surface less adhesive, the master was treated with trichloro(1H,1H,2H,2H-perfluorooctyl)silane vapor. Sylgard 184 (Dow Corning) base and curing agent were mixed in a 10:1 ratio by mass and degassed. After pouring over the master, the PDMS was cured for 1 hr at 80°C (*Figure 1—figure supplement 2*). Features were confirmed with helium ion microscopy and scanning electron microscopy. The PDMS was then removed from the mask and baked for an additional 1 week at 80°C. To extract low molecular weight species that diffuse to the surface and alter surface chemistry (*Eddington et al., 2006*), PDMS was incubated, in sequence, with triethylamine, ethyl acetate, and acetone and allowed to dry overnight in an oven.

## MT bridges

Silanization of PDMS was similar to glass with a few modifications. Ethanol was used as the solvent rather than acetone, as it is less likely to swell the PDMS (*Lee et al., 2003*). The patterned surface was plasma oxidized at 50 W and 200 mTorr for 1 min. The slab was then immediately immersed in a 5% (v/v) solution of APTES in HPLC-grade ethanol (Sigma, 459828) for 20 min. After rinsing in 95% ethanol/water, the PDMS was baked 40 min on a hot plate. To remove unbound silane, PDMS was rinsed in pure ethanol and then water, and baked for an additional 40 min. Finally, PDMS was incubated for 1 hr in an 8% glutaraldehyde solution. Excess glutaraldehyde was removed by rinsing in water and the functionalized PDMS was stored at room temperature for 1 week. To create a flow chamber, uncured PDMS was spin-coated on a coverslip to a thickness of 100 µm. After baking, a channel was cut in the PDMS-coated coverslip, and the surface was plasma oxidized. The PDMS block with the bridge pattern was placed functional-side down on the coverslip, creating a flow cell (*Figure 1—figure supplement 2*). Motility assays were performed as described for single-molecule assays. The sample was imaged using a 1.4 NA oil-immersion condenser (Nikon) under brightfield illumination.

## Anti-GFP coating beads

Latex beads were coated with anti-GFP antibody as described (*Belyy et al., 2016*). To prevent clumping of the beads when incubated with high concentrations of motors, we used 'CML' beads (ThermoFisher, C37481), which have a high density of carboxyl groups that facilitate charge repulsion. 200 µL of 0.5 µm diameter CML beads (4% solids) were washed three times in activation buffer (10 mM MES, 100 mM sodium chloride, pH 6.0) by centrifugation for 6 min at 7,000 *g*. Final resuspension was in 200 µL activation buffer. Beads were then sonicated for 1 min in a bath sonicator (Vevor). Separately, fresh 4 mg/mL solutions of EDC (1-ethyl-3-(3-dimethyl aminopropyl)carbodiimide hydrochloride, ThermoFisher) and Sulfo-NHS (N-hydroxysulfosuccinimide) were prepared in water. 5 µL each of fresh EDC and Sulfo-NHS solutions were added to the beads. Beads were sonicated for 1 min and nutated for 30 min at room temperature. After three washes in PBS, beads were

resuspended in 200 µL PBS and mixed with 200 µL 0.4 mg/mL anti-GFP antibody overnight with nutation. Custom-made anti-GFP antibodies (Covance) were purified by GFP affinity chromatography. The beads were passivated by incubating with 10 mg/mL bovine serum albumin (BSA) overnight. Finally, beads were washed five times in PBS and stored at 4°C with 1 mg/mL BSA supplement.

## Bead motility assay

For bead motility assays, anti-GFP beads were diluted two-fold in water and sonicated for 1 min to disperse the beads. GFP-tagged kinesin motors were incubated with beads for 10 min. The excess motor was then washed from the beads by diluting the mixture into 100 µL BRB80 supplemented with 1 mg/ml casein (BRB-C) and centrifuging at 8,000 $g$ for 3 min. The supernatant was removed and the pellet was resuspended in 15 µL BRB-C supplemented with 1 mM TCEP, 100 µM ATP, glucose oxidase, catalase, and 0.4% dextrose. Beads were then flown into a flow chamber after surface-immobilization of biotinylated MTs. DDB experiments were performed similarly with a few exceptions. The GFP handle was on the cargo adaptor (BicD2N-GFP). We used a dynein mutant that does not form the autoinhibited phi-conformation (*Zhang et al., 2017*) to facilitate assembly of the DDB complex. 1 µL each of 1 µM human dynein complex, pig brain dynactin, and BicD2N-GFP were mixed at a 1:1:1 molar ratio and incubated for 15 min before mixing with the beads. The mixture was pelleted at 8,000 $g$, resuspended in 15 µL MB supplemented with 1 mg/mL casein, 1 mM TCEP, 100 µM ATP, glucose oxidase, catalase and dextrose and added to the flow chamber.

## Statistical analysis

Each measurement was performed with at least three independent replicates, and the exact number of repetitions is reported for each experiment. Each statistical analysis method is explicitly stated in the main text and/or figure legend. 'n' refers to the number of motors analyzed across all experimental replicates. 'Independent experiments' mean data collected on different days using the same protein preparation. However, all major findings were repeated with multiple different protein preparations. Standard deviations (SD) represent differences between independent experiments. When reported, standard error (SEM) refers to the error of the fit.

## Data and materials availability

Data has been deposited in Dryad Digital Repository and can be currently accessed at https://doi.org/10.6078/D1P09W.

# Acknowledgements

We are grateful to V Belyy, A Jack and Y Ezber for helpful discussions, SM Luk for helium ion microscopy, P Lum and N Azgui at the Biomolecular Nanotechnology Center for help with nanofabrication, and A Killilea for the mammalian cell culture. This work was funded by grants from the NIH (GM094522), and NSF (MCB-1055017, MCB-1617028) to AY and a grant from the NIH (5 F31 GM123655-03) to LF

# Additional information

### Funding

| Funder | Grant reference number | Author |
| --- | --- | --- |
| National Institute of General Medical Sciences | GM094522 | Ahmet Yildiz |
| National Science Foundation | MCB-1055017 | Ahmet Yildiz |
| National Science Foundation | MCB-1617028 | Ahmet Yildiz |
| National Institute of General Medical Sciences | GM123655-03 | Luke S Ferro |

The funders had no role in study design, data collection and interpretation, or the decision to submit the work for publication.

### Author contributions
Luke S Ferro, Conceptualization, Data curation, Software, Formal analysis, Validation, Investigation, Methodology, Writing—original draft; Sinan Can, Resources, Software, Validation, Methodology; Meghan A Turner, Conceptualization, Data curation, Formal analysis; Mohamed M ElShenawy, Data curation, Software; Ahmet Yildiz, Conceptualization, Supervision, Funding acquisition, Validation, Investigation, Writing—original draft, Writing—review and editing

### Author ORCIDs
Luke S Ferro (ID) https://orcid.org/0000-0001-5037-8464
Ahmet Yildiz (ID) https://orcid.org/0000-0003-4792-174X

### Decision letter and Author response
Decision letter https://doi.org/10.7554/eLife.48629.027
Author response https://doi.org/10.7554/eLife.48629.028

## Additional files

### Supplementary files
• Transparent reporting form
DOI: https://doi.org/10.7554/eLife.48629.023

### Data availability
Data has been deposited in Dryad Digital Repository and can be currently accessed at https://doi.org/10.6078/D1P09W.

The following dataset was generated:

| Author(s) | Year | Dataset title | Dataset URL | Database and Identifier |
|---|---|---|---|---|
| Luke S Ferro, Sinan Can, Meghan A Turner, Mohamed M ElShenawy, Ahmet Yildiz | 2019 | Data from: Kinesin and dynein use distinct mechanisms to bypass obstacles | https://doi.org/10.6078/D1P09W | Dryad Digital Repository, 10.6078/D1P09W |

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
