## [Decision Letter]

Thank you for submitting your article "Kinesin and dynein use distinct mechanisms to bypass obstacles" for consideration by *eLife*. Your article has been reviewed by three peer reviewers, one of whom is a member of our Board of Reviewing Editors, and the evaluation has been overseen by Vivek Malhotra as the Senior Editor. The following individual involved in review of your submission has agreed to reveal their identity: Robert Anthony Cross (Reviewer #2).

The reviewers have discussed the reviews with one another and the Reviewing Editor has drafted this decision to help you prepare a revised submission.

Summary:

This is an interesting study comparing the abilities of cytoplasmic dynein and kinesin-1 to step around large obstacles attached to the microtubule lattice. Previous roadblock experiments, of which there are many, did not use a full length dynein in complex with BicD and dynactin. Also, the obstacles used here are larger than those used in any previous study (25nm diameter quantum dots (QD), or a bar of PDMS). The main result is that single dynein-BicD molecules, and to a lesser extent single yeast dynein molecules, can circumvent large obstacles. A subsidiary result is that beads carrying multiple kinesins can also move around obstacles, whereas single kinesin molecules cannot.

All reviewers found this study interesting, but they raised the following major concerns that need to be addressed for you to submit a revised manuscript:

Essential revisions:

1) Controls are required to address the question to which extent the results are specific for the combination of large obstacle and fluorescent label used here (QDs). Looking at the literature, there seems to be a tendency that larger obstacles like quantum dots may hinder motor movement more than proteins. For controls, motors could be labelled with a small fluorophore which could be attached to the Halotag and/or smaller proteins could be used as obstacles (e.g. rigor kinesins or streptavidins).

2) The 'stuck' state observed for kinesin is quite different from what most studies have reported for single kinesins with smaller dyes encountering smaller obstacles. The same controls suggested above could shed light on this discrepancy with the literature.

3) In discussing the circumvention of obstacles, 'rotation' of the motor-bead is used in two different senses and the distinction is not made clearly. Migration of the motor-bead complex around the MT is one kind of rotation. Rotation of the multimotor-bead around its own centre (rolling) as it does this is another kind of rotation. Directional rolling as the bead migrates around the MT axis would indicate that new motors are being recruited into the interface with the lattice. Alternatively, motors might just maintain diffusional attachment to the MT by marching on the spot until the team drifts sideways passively around the obstruction, without generating sideways force impulses? Is there evidence that sideways motion across the lattice is "driven by forces exerted by the motors"? These questions deserve more discussion.

4) Some technical clarifications should be provided:

4a) It was unclear how the authors know that their kinesin-QD complexes correspond to single kinesin molecules. Are there measurements on the GFP fluorescence versus the QD fluorescence? Run length might give reassurance that these complexes are single molecules. Data in Figure 1—figure supplement 2 show run length above 2 µm, which seems little long for a single molecule? If there are fluorescence ratio measurements, can an estimate be made of the number of kinesins needed to make a multimolecule team capable of circumventing obstacles?

4b) Please explain in more detail how the number of QDs on a microtubule was calculated from the intensity of the QD spots?

4c) Figure 2C: It seems that n means the number of samples scored. Was the number of replicates only one (meaning, both the protein preparation and the assay were performed only once for each data point in the figure)? Has the standard error been derived from regression analysis that was applied to a single data set? Please explain the statistical details.

---

## [Author Response]

Essential revisions:1) Controls are required to address the question to which extent the results are specific for the combination of large obstacle and fluorescent label used here (QDs). Looking at the literature, there seems to be a tendency that larger obstacles like quantum dots may hinder motor movement more than proteins. For controls, motors could be labelled with a small fluorophore which could be attached to the Halotag and/or smaller proteins could be used as obstacles (e.g. rigor kinesins or streptavidins).2) The 'stuck' state observed for kinesin is quite different from what most studies have reported for single kinesins with smaller dyes encountering smaller obstacles. The same controls suggested above could shed light on this discrepancy with the literature.

To address these concerns, we collected new experimental data. We labeled motors with fluorescent dyes and decorated MTs with tubulin antibody obstacles. As predicted by the reviewers, antibody obstacles inhibit kinesin motility by causing them to detach from MT rather than getting stuck. These data are presented in a new Figure 3. and the results are discussed in the main text as follows:

“Kinesin quickly detaches from MTs decorated with antibody obstacles

Previous studies have reported that kinesin motors detach when encountering catalytically “dead” motors or MAPs on the MT rather than getting stuck (Telley et al., 2009, Schneider et al., 2015). […] However, kinesin motors that carry a rigid QD cargo are more likely to pause when encountering a bulkier obstacle on an MT.”

3) In discussing the circumvention of obstacles, 'rotation' of the motor-bead is used in two different senses and the distinction is not made clearly. Migration of the motor-bead complex around the MT is one kind of rotation. Rotation of the multimotor-bead around its own centre (rolling) as it does this is another kind of rotation. Directional rolling as the bead migrates around the MT axis would indicate that new motors are being recruited into the interface with the lattice. Alternatively, motors might just maintain diffusional attachment to the MT by marching on the spot until the team drifts sideways passively around the obstruction, without generating sideways force impulses? Is there evidence that sideways motion across the lattice is "driven by forces exerted by the motors"? These questions deserve more discussion.

We would like to thank the reviewers for making a distinction between these possibilities. We did not study the mechanism by which the beads move from one side of an MT to the other side on PDMS walls. Therefore, we cannot distinguish between the possibilities of “rolling of the bead” or sideways stepping of the motors bound to the bead. We revised the Discussion to mention these possibilities. We have also mentioned forced detachment of motors from MT as a possibility, rather than as a fact, in Discussion of the revised manuscript.

We also noted that the rotation of the beads around the PDMS wall is different than the helical movement of the beads. In the former, the beads do not move along the MT long axis until they move to the top of the wall, whereas in the latter, the majority of the movement occurs in the longitudinal direction. We wrote: "This movement was different from the previously observed helical movement of kinesin- or dynein-driven cargos around the MT, in which rotation is accompanied by forward translational movement."

4) Some technical clarifications should be provided:4a) It was unclear how the authors know that their kinesin-QD complexes correspond to single kinesin molecules. Are there measurements on the GFP fluorescence versus the QD fluorescence? Run length might give reassurance that these complexes are single molecules. Data in Figure 1—figure supplement 2 show run length above 2 µm, which seems little long for a single molecule? If there are fluorescence ratio measurements, can an estimate be made of the number of kinesins needed to make a multimolecule team capable of circumventing obstacles?

To make our measurement more consistent with the field, we reported the half-life of the exponential decay (rather than the decay constant) as the motor run length in the revised manuscript. The corrected run length for kinesin-QD in Figure 1—figure supplement 1 is 2.5 ± 0.4 µm (mean ± SD).

To determine if this run length is abnormally high, we measured the run lengths of kinesin motors labeled with LD555 (Lumidyne) dye. This dye is directly conjugated to Trolox, which increases its photostability. The run length of kinesin-LD555 (1.95 ± 0.03 µm, mean ± SD) is not significantly different from QD-tagged motors (p = 0.21, two-tailed t-test). To prevent attachment of multiple kinesins to a single QD, we always kept QDs in excess of motors during the labeling reactions. In Materials and methods, we wrote: “2 µM of these QDs were mixed with 100-500 nM motors fused with a SNAPTag or HaloTag for 10 min on ice.” Therefore, it is unlikely that the QDs are driven by multiple motors.

4b) Please explain in more detail how the number of QDs on a microtubule was calculated from the intensity of the QD spots?

This information was added to the “Motility” section in the Materials and methods: “At low QD concentrations, the linear density of QD-obstacles on the MT was measured by counting the number of fluorescent spots from TIRF images. At higher QD concentrations, the linear density was estimated from the ratio of total fluorescence signal on an MT to the fluorescence intensity of a single QD."

4c) Figure 2C: It seems that n means the number of samples scored. Was the number of replicates only one (meaning, both the protein preparation and the assay were performed only once for each data point in the figure)? Has the standard error been derived from regression analysis that was applied to a single data set? Please explain the statistical details.

An explanation has been included in Materials and methods: ““n” refers to the number of motors analyzed across all experimental replicates. […] When reported, standard error (SEM) refers to the error of the derived parameters from a fit.”